# Semaphorin 3C as a Therapeutic Target in Prostate and Other Cancers

**DOI:** 10.3390/ijms20030774

**Published:** 2019-02-12

**Authors:** Daniel H.F. Hui, Kevin J. Tam, Ivy Z.F. Jiao, Christopher J. Ong

**Affiliations:** Vancouver Prostate Centre and Department of Urologic Sciences, University of British Columbia, Vancouver, BC V6H 3Z6, Canada; danielhui78@gmail.com (D.H.F.H.); kjtam8@gmail.com (K.J.T.); ijiao@prostatecentre.com (I.Z.F.J.)

**Keywords:** semaphorins, SEMA3C, inhibitors, plexins, neuropilins, cancer therapeutics

## Abstract

The semaphorins represent a large family of signaling molecules with crucial roles in neuronal and cardiac development. While normal semaphorin function pertains largely to development, their involvement in malignancy is becoming increasingly evident. One member, Semaphorin 3C (SEMA3C), has been shown to drive a number of oncogenic programs, correlate inversely with cancer prognosis, and promote the progression of multiple different cancer types. This report surveys the body of knowledge surrounding SEMA3C as a therapeutic target in cancer. In particular, we summarize SEMA3C’s role as an autocrine andromedin in prostate cancer growth and survival and provide an overview of other cancer types that SEMA3C has been implicated in including pancreas, brain, breast, and stomach. We also propose molecular strategies that could potentially be deployed against SEMA3C as anticancer agents such as biologics, small molecules, monoclonal antibodies and antisense oligonucleotides. Finally, we discuss important considerations for the inhibition of SEMA3C as a cancer therapeutic agent.

## 1. Introduction

The semaphorins constitute a broadly distributed and functionally diverse family of signalling proteins. While the semaphorins are best known for their roles in development and neuronal outgrowth, their involvement in malignancy is becoming increasingly appreciated. Given the versatile nature of the semaphorins, it comes as little surprise that the semaphorins have been implicated in numerous different oncogenic processes. While reviews of the semaphorins [1], semaphorins in cancer [2,3,4], and semaphorins as targets of therapy [5] are available and detailed elsewhere, this report surveys the potential in targeting SEMA3C in cancer. This overview builds on the knowledge of the normal and oncogenic roles of SEMA3C which were recently summarized [6].

## 2. Discovery

The first evidence for the existence of the semaphorins was uncovered in the early 1990s by Raper & Kaplfhammer [7] who showed that a component of chick brain extracts had the capacity to trigger dorsal root ganglion collapse. Although the identity of that protein was not yet known, this work ushered in the seminal work by Kolodkin and colleagues who isolated and identified the first semaphorin in 1992, at that time referred to as Fascilin IV (now known as SEMA1A) from grasshopper extract [8]. In this work, using Fascilin IV-neutralizing monoclonal antibodies, Fascilin IV was shown to be essential for proper axon extension in the grasshopper embryo. In the wake of this finding, additional work added members to the class of proteins now known as semaphorins, including the discovery of the first vertebrate semaphorin in 1993 by Luo et al. [9] and SEMA3C by Püschel et al. in 1995 [10]. While pioneering work on SEMA3C identified it as a repelling factor in neurite extension, subsequent studies noted the embryonic lethal phenotype in SEMA3C knockout mice and its responsibility for cardiovascular development [11]. Shortly after the seminal work on SEMA3C in development in the early 1990s, recognition of SEMA3C for its roles of in carcinogenesis began to surface in a variety of different cancer types beginning with ovarian and lung cancer [12,13,14]. The list of cancers that SEMA3C is implicated in has steadily grown and now includes gastric, lung, liver, breast, gynecological, prostate, pancreatic and brain (Figure 1) [12,13,14,15,16,17,18,19,20,21,22,23,24,25,26]. SEMA3C continues to receive attention for both its roles in development and for its involvement in cancer biology. 

## 3. Structure & Function

The semaphorins are phylogenetically related proteins consisting of over twenty members that fall into one of eight different classes distinguished from one another by different molecular features such as immunoglobulin (Ig) domains, basic domains, thrombospondin repeats or glycosylphosphatidylinositol (GPI) linkages [3]. All semaphorins, however, have in common a 500 amino acid N-terminal semaphorin domain which assumes 7-bladed β-propeller topology. Classes 1, 2 and 5 are found in invertebrates while classes 3 through 7 are found in vertebrates. The eighth class is found in viruses (Class V). Semaphorins can be tethered to the cell through membrane-spanning regions or GPI anchors, or can be secreted by the cell. Of the vertebrate semaphorins, the class 3 semaphorins are secreted. Molecular territories delineated by secreted semaphorins provide directional cues for cell movement which links SEMA3C to invasion & metastasis—one of the key hallmarks of cancer [27]. Semaphorin signalling is transduced across the plasma membrane by plexin (PLXN) receptors which possess intrinsic GAP (GTPase activating protein) activity [28,29,30,31] and associate with intracellular guanine nucleotide exchange factor (GEF) and GAP effector proteins but can also transactivate receptor tyrosine kinases [32,33]. With the exception of SEMA3E, all class-3 semaphorins require neuropilin (NRP) coreceptors to bind plexins [34,35], but recent work by Smolkin et al. indicates that SEMA3C can function through PLXNA4 and PLXND1 in the absence of neuropilins [36]. Both plexins and neuropilins are also heavily discussed in the field of cancer biology. The receptors to SEMA3C are classically regarded as NRP1, NRP2, PLXNA2, PLXNA4, PLXNB1, PLXND1, and possibly PLXNA1 [37,38,39,40]. The fact that there are multiple receptors for SEMA3C together with the fact that certain neuropilin and plexin members are shared by multiple semaphorins collectively underscore the intricacy of semaphorin signalling but also foreshadow potential challenges in targeting all of, but only, the intended semaphorin axis—the oncogenic programs of SEMA3C in the case of this report. Broadly speaking, the receptors to semaphorins are known to cross-talk with receptor tyrosine kinases (RTKs) such as vascular endothelial growth factor receptor 2 (VEGFR2), human epidermal growth factor receptor 2 (HER2), epidermal growth factor receptor (EGFR), hepatocyte growth factor receptor (MET), and off-track kinase (OTK) [3,39]. The cellular activities downstream of these RTKs are broad so strong specificity of inhibitors for SEMA3C will be paramount in order to mitigate off-target effects.

Outside of cancer, whilst primarily noted for their participation in the development of the nervous system [10,41,42,43,44,45,46,47,48,49], research has also established the importance of SEMA3C and its receptors in cardiovascular [11,50,51,52,53,54,55,56], retinal [57] and renal [58] development, as well as chondrogenesis [59], and alveolar growth and repair [60]. In addition, SEMA3C has been associated with other human conditions including autism [61], Takao syndrome [62], Alzheimer’s [63] and Hirschsprung disease [64].

## 4. SEMA3C by Cancer Type

The different oncogenic processes that SEMA3C is involved in have been reviewed previously [6]; the following discussion is meant to build on those findings by highlighting the different cancer types that SEMA3C is implicated in. Potential molecular strategies to achieve SEMA3C inhibition are then considered.

### 4.1. Prostate Cancer

#### 4.1.1. SEMA3C as the First Bona Fide Prostate Cancer Andromedin

Prostate cancer (PCa) is the most prevalent noncutaneous cancer in males and is the second leading cause of cancer related deaths of men in North America [65]. Androgen deprivation therapy (ADT) is currently the first-line systemic treatment for advanced, metastatic PCa. The exquisite dependency of PCa on androgens for growth and survival was first recognized in the 1940’s when Huggins and Hodges demonstrated the antitumor activity of hormonal manipulation in the treatment of PCa [66]. Since then, ADT has been the standard of care in the treatment of metastatic and locally advanced PCa. Drugs targeting the androgen/androgen receptor axis have been well-validated clinically and remain without a doubt the most effective class of therapies for treatment of advanced PCa (Figure 2). The androgen receptor (AR) is a hormone-dependent transcription factor and AR signaling primarily follows the classical mechanism of nuclear receptor signaling [67]. Upon androgen binding to the AR, the androgen–AR complex is activated and functions as a transcription factor that regulates the expression of a number of AR target genes. The ultimate goal of therapeutic AR pathway inhibition is primarily to inhibit the transcriptional output of the AR. Despite the central role of AR pathway in PCa biology, the nature of these androgen-regulated genes that drive PCa growth/survival has been relatively poorly elucidated. 

A first clue regarding the androgen-regulated factors that mediate growth and survival came from the Cunha laboratory in the early 1970’s who showed from tissue recombination studies that prostate development was dependent on reciprocal interactions between the epithelium and the mesenchyme of the urogenital sinus [68,69,70]. They discovered that hormonal effects on the epithelium were mediated by secreted soluble paracrine factors produced by mesenchymal/stromal cells in an androgen-regulated manner. These findings naturally spawned the “andromedin hypothesis” which posits that the paracrine mediators could be androgen-mediated growth factors called andromedins. Andromedins are thought to diffuse from the stroma into the epithelial layers and orchestrate growth and differentiation of the prostate by binding to cognate receptors on epithelial cells [71] (Figure 3). Over the years, a number of growth factors have been implicated as andromedins such as fibroblast growth factor 7 (FGF7), fibroblast growth factor 10 (FGF10), and insulin-like growth factor 1 (IGF1) [72,73,74]. However, since none of these are androgen-regulated, a true andromedin has remained elusive. In early 2000s, seminal work by Issacs found that the malignant transformation of normal prostatic epithelial cells is associated with a switch from a paracrine to an autocrine mechanism in androgen-stimulated growth [75,76]. Whether the andromedins that play a role in normal prostate development and in cancer are the same or different is currently under investigation [77].

We have recently found that SEMA3C drives cancer growth by transactivating multiple receptor tyrosine kinases including EGFR, HER2 and MET via Plexin B1 [39] (Figure 4). Furthermore, we found that SEMA3C is an androgen-induced gene with an AR-induced enhancer containing an androgen response element in intron 2 of SEMA3C [78]. Interestingly, this AR enhancer was also independently identified to be among the top 10 AR-induced enhancers from an unbiased genome-wide functional screen of AR binding sites (personal communication, Dr. Nathan Lack, University of British Columbia). Notably, we found that SEMA3C is a secreted, soluble autocrine growth factor in PCa and importantly combined with our findings that SEMA3C is transcriptionally regulated by AR in a GATA-binding protein 2 (GATA2)-dependent manner [78], these data collectively make SEMA3C the first bona fide PCa-derived andromedin to be identified. The identification of SEMA3C as an androgen-induced autocrine growth factor in PCa makes SEMA3C a promising new target for treatment of metastatic castration-resistant prostate cancer (mCRPC).

Despite the initially high response rate to ADT, lethal mCRPC emerges almost universally over time in men with metastatic disease. Treatment of mCRPC with more potent next generation AR pathway inhibitors (ARPIs) such as abiraterone and enzalutamide have demonstrated meaningful clinical responses in mCRPC patients. However, clinical responses are variable and relatively short-lived as mCRPC tumors almost always acquire resistance to second-generation hormonal therapies. Since the introduction of ARPIs, there has been an observable shift in the clinical phenotype of mCRPC with increased numbers of patients developing neuroendocrine PCa (NEPC) and double negative PCa [79]. However, it is important to note that the majority (over 60%) of mCRPC that progress post-ARPI remain AR driven. 

#### 4.1.2. AR Reactivation Drives SEMA3C-Induced Growth

Multiple mechanisms underlie PCa progression despite castrate levels of androgens in blood [80]. The most common castration-resistance mechanism is through reactivation of AR signaling via multiple pathways including: AR overexpression and AR gene amplification, AR mutations leading to broader ligand specificity, intratumoral androgen biosynthesis, and AR variants that are constitutively active in absence of ligand [80] (Figure 5). Since SEMA3C drives RTK pathway activation, regulation of SEMA3C by AR functionally links AR to cell growth and survival pathways such as RTK activation and downstream rat sarcoma viral small GTPase oncogene (RAS)/ mitogen activated protein kinase (MAPK) and phosphoinositide 3-kinase (PI3K)/ protein kinase B (AKT) signaling pathways. Moreover, since SEMA3C is an AR-driven gene, this AR reactivated subset of mCRPC may benefit from co-targeting of ADT and ARPI with SEMA3C inhibitors.

#### 4.1.3. SEMA3C and Tumor cell growth via RTK/AR Crosstalk and AR Bypass 

Although AR is a central player in mCRPC, other mechanisms governing development of castration-resistance have been proposed such as cross talk between AR and RTK signaling and non-AR-mediated bypass mechanisms (Figure 5). Signaling via RTKs such HER2 and EGFR has been shown to activate AR in ligand independent manner [80]. Since SEMA3C can drive activation of multiple receptor tyrosine kinases including HER2 and EGFR [39], SEMA3C may promote castrate resistance growth via AR/RTK pathway crosstalk. An alternate mechanism for achieving castration resistance is through the bypass mechanism where alternative pathways are activated that bypass the requirement for androgen/AR axis for mediating growth and survival [80]. SEMA3C expression was found to be negatively regulated by the AR pioneering factor forkhead box A1 (FOXA1) and consequently, loss of FOXA1, a gene that is mutated in a subset of PCa, leads to constitutive high SEMA3C expression [78]. Furthermore, SEMA3C overexpression was found to promote castrate-resistant PCa growth in vitro and in vivo [39], suggesting that FOXA1 mutations may confer castrate-resistant growth via bypass mechanisms through constitutive SEMA3C expression. 

#### 4.1.4. SEMA3C and Cancer Stem Cells

As alluded to earlier, SEMA3C is thought to contribute to the progression of PCa by promoting cancer recurrence. Another proposed mechanism of resistance to ADT is the presence of cancer stem cells (CSCs) (Figure 5). The theory that CSCs within a tumor, which represent only a small subset of the tumor, is responsible for its growth was first supported by evidence in acute myeloid leukemia [81] followed by the first evidence of CSCs in solid tumors in breast cancer [82]. The concept of CSCs was then extrapolated to other cancers, including PCa [83]. In PCa, prostate CSCs are postulated to undergo androgen independent growth/survival and to persist after ADT, and seed tumor relapse as mCRPC [84,85]. Overexpression of SEMA3C in prostate cells promotes a more stem-like phenotype, characterized by an increased number of cells expressing the prostate stem cell marker cluster of differentiation-44 (CD44) as well as improved sphere formation [86]. The concept of targeting CSCs in PCa to improve therapeutic success and prevent relapse is not novel, but identifying appropriate molecular targets to eliminate CSCs remains a significant challenge [87,88]. Since SEMA3C promotes development of a stem-like phenotype, a state which has been proposed to mediate metastasis [89], inhibitors of SEMA3C signalling represent a potential novel therapeutic approach in metastatic CRPC in which the therapeutic options are currently limited.

#### 4.1.5. SEMA3C and Epithelial-to-Mesenchymal Transition

Epithelial-to-mesenchymal transition (EMT) is an important process converting compact and ordered epithelial cells into dispersed migratory mesenchymal cells. In normal physiology, neural crest precursors at the dorsal aspect of the neural tube undergo EMT to become highly migratory neural crest cells that migrate extensively to diverse locations to form many specialized structures and tissues in the developing embryo including melanocytes and cardiac neural crest cells. Given the molecular and cellular similarities between pathological and developmental EMTs, studying the EMT process during neural crest development may give insights into the process of EMT and metastasis in cancer. In normal development, SEMA3C is thought to play a role in development and migration of cardiac neural crest cells. Mice lacking *Sema3c* exhibited interruption of the aortic arch and persistent truncus arteriosis as well as defects in migration of cardiac neural crest cells towards the outflow tract [90]. Interestingly, in some animals, heart defects were accompanied by ectopic pigmentation in the heart, lung and other tissues, and hypopigmentation of the skin suggesting that SEMA3C also plays a role in differentiation and migration of neural crest-derived melanocytes [90]. Epithelial prostate cells overexpressing SEMA3C lose their cobblestone architecture and exhibit a spindle-like appearance. In line with these phenotypic changes, these cells express more mesenchymal markers such as N-cadherin and fibronectin and show increased incidence of metastases when injected into mice [86]. The EMT induced by SEMA3C may promote metastatic potential of prostate tumors. The link between SEMA3C and EMT and cancer stem cells punctuates the importance in exploring SEMA3C or its receptors as potential cancer targets.

#### 4.1.6. SEMA3C and RTK Coactivation

RTKs are central to many processes in cancer and targeted anti-RTK therapies have shown clinical success in treatment of numerous cancers. Recently, simultaneous activation of multiple RTKs referred to as RTK co-activation is becoming increasingly recognized as an important feature in many cancers [91]. In fact, RTK’s are rarely found to act alone but rather, they typically act as networks of multiple RTKs that cooperate and transmit coordinated and highly integrated signals. Multiple crosstalk mechanisms leading to activation of multiple RTKs have been proposed. In the absence of RTK gene mutations leading to constitutive receptor activation, it is assumed that cognate ligands play a crucial role in autocrine or paracrine stimulation of these RTK pathways. SEMA3C is a secreted soluble factor that can simultaneously transactivate multiple RTK pathways in a cognate ligand-independent manner. The concept of RTK co-activation has major implications in predicting tumor responses to targeted therapeutics and chemoresistance mechanisms. In PCa, single agents targeting individual RTK pathways have failed to show meaningful clinical responses despite clear evidence of pathway inactivation. Since multiple RTK pathways are activated in PCa by SEMA3C, it is not surprising that targeting single RTKs individually would be ineffective due to redundancy of bypass RTK pathways and could explain intrinsic resistance of PCa to targeted RTK therapies such as EGFR inhibitors (erlotinib, gefitinib) as well as anti-HER2-targeted antibody therapeutics (pertuzumab, trastuzumab) [92,93].

Similar to SEMA3C’s role in mediating intrinsic resistance of PCa to targeted RTK therapies, SEMA3C may also play a role in facilitating acquired resistance of cancer to RTK targeted agents. A common mechanism mediating acquired resistance to RTK inhibition and/or tyrosine kinase inhibitors (TKIs) is activation of secondary RTK pathways that create a bypass track [94]. For example, resistance to anti-EGFR monoclonal antibodies in colorectal cancer and to EGFR TKIs in EGFR-mutant non-small cell lung cancer (NSCLC) can be mediated by activation of alternate RTK pathways including MET and HER2 [94]. How cancer cells switch from one RTK pathway to another is assumed to require upregulation of both the secondary RTK and its cognate ligand. Thus, the ability of SEMA3C to simultaneously transactivate multiple RTKs such as EGFR, HER2 and MET could facilitate the switch of primary dependency of cancer growth from one RTK pathway to another. Thus, it is interesting to postulate whether SEMA3C’s ability to coordinately activate multiple RTK pathways may play a role in the setting of acquired resistance to RTK-targeted therapies in lung, head and neck, breast, colon and other cancers.

### 4.2. Role of SEMA3C in Other Cancers

SEMA3C and its receptors continue to draw great attention in the context of numerous cancer [6]. Among the class 3 semaphorins, SEMA3C is notable because its expression is most consistently associated with poor prognosis in a wide spectrum of cancers (Figure 1). High SEMA3C expression is associated with unfavourable outcomes in glioma, breast, lung, liver, pancreatic, gastric, gynecological, and prostate cancers [12,13,14,15,16,17,18,19,20,21,22,23,24,25,26]. Thus, given SEMA3C’s ability to activate multiple RTK pathways and its key role in prostate cancer growth and survival, it is intriguing to speculate whether SEMA3C might also play an important role in driving angiogenesis, cell growth, cell survival and metastasis in other cancers.

#### 4.2.1. Pancreatic Cancer

Pancreatic cancer has one of the lowest five-year survival rates of all cancers and is the fourth leading cause of cancer deaths in both men and women [65]. Many of these patients are diagnosed with late-stage disease for which few treatment options are available. 95% of patients with pancreatic ductal adenocarcinoma (PDAC), the most common pancreatic malignancy, contain a mutation in the Kirsten rat sarcoma oncogene (*KRAS*) [95,96]. KRAS plays roles in both the initiation and maintenance of pancreatic adenocarcinoma by altering metabolic pathways such as increasing glucose uptake and autophagy [97,98]. The role of *KRAS* mutations in promoting tumor growth and its prevalence in PDAC have made KRAS an attractive therapeutic target. However, despite continuing efforts, an effective anti-RAS treatment has yet to be discovered, underlining the need for alternate strategies to target this pathway [99,100].

In 2018, Xu et al. found multiple protumoral roles for SEMA3C in pancreatic cancer. SEMA3C is upregulated in pancreatic tumor tissue compared to normal tissue, and SEMA3C expression within pancreatic tumors is associated with a more advanced TNM classification of malignant tumors (TNM) stage and decreased one-year survival [19]. Overexpression of SEMA3C results in increased proliferation, suppressed apoptosis, increased invasion, and increased tumor volume which are mediated through increased extracellular signal-regulated kinase (ERK) signalling. This is congruent with previous evidence showing that ERK activation in pancreatic cancer promotes both proliferation and EMT [101,102].

One current area of exploration to inhibit KRAS pathway activation is to indirectly target its downstream effector pathways [100,103] which include the PI3K/AKT pathway and the rapidly accelerated fibrosarcoma serine/threonine kinase (Raf)/ MAPK kinase (MEK)/ERK pathway. The PI3K/AKT and Raf/MEK/ERK pathways are commonly associated with cancer due to their roles in the regulation of diverse cell functions including proliferation and metabolism. A mutation leading to constitutive activation of the PI3K pathway induced PDAC formation which phenocopied lesions resulting from KRAS mutations [104]. Furthermore, treatment with a class I PI3K inhibitor was able to block tumor growth in KRAS-driven PDAC [104]. MEK1/2 inhibitors had similar inhibitory effects on PDAC resulting from KRAS mutation [105]. Knockout of EGFR also suppresses growth of tumors in KRAS^G12D^ mice, which is thought to be due to insufficient ERK activation [106,107]. This would suggest that these pathways are necessary for the oncogenic effects of *KRAS* mutations and suppression of these pathways could prevent tumorigenesis in this context. 

There is considerable overlap between the effector pathways of SEMA3C and KRAS; both lead to PI3K/AKT and Raf/MEK/ERK activation and SEMA3C expression positively correlates with *KRAS* mutations in The Cancer Genome Atlas (TCGA) datasets of pancreatic adenocarcinoma [108,109]. Taken together, this raises the possibility that SEMA3C signalling may have a role in maintaining the oncogenic effects of *KRAS* mutations in pancreatic cancer. In line with this, preliminary unpublished work in our laboratory has shown that attenuation of SEMA3C signalling dampens cell proliferation and oncogenic cell signalling in panel of KRAS mutant pancreatic cancer cell lines. SEMA3C may also be involved in promoting resistance of pancreatic tumors to gemcitabine, a nucleoside analog used as a chemotherapy medication in pancreatic cancer. Given that SEMA3C has been demonstrated to drive stemness in other cancers [6], and the fact that cancer stem cells in pancreatic tumors are thought to be responsible for chemoresistance [110], it would be interesting to explore whether SEMA3C’s ability to induce stemness extends to the context of pancreatic cancer.

#### 4.2.2. Brain Cancer

SEMA3C signalling is hypothesized to be involved in promoting glioma malignancy based on the observation that human glioma cell lines express high levels of SEMA3C and its receptors [20]. SEMA3C correlates with the severity of glioma: SEMA3C expression is markedly increased in grade IV human glioma tumor samples (glioblastomas) compared to grades I-III glioma samples and higher expression levels of SEMA3C were associated with poorer survival rate [111].

Glioblastomas are very difficult to treat, and glioma stem cells have been identified as one of the major contributors to tumorigenesis and therapeutic resistance. Glioma stem cells can promote angiogenesis through overexpression of vascular endothelial growth factor (VEGF), particularly under hypoxic conditions [112]. In addition, glioma stem cells show an increased ability to repair DNA damage and promote growth after therapy leading to increased resistance [113,114]. Therefore, glioma stem cells have been identified as attractive therapeutic targets in glioblastoma [115,116].

The mechanism through which SEMA3C promotes glioma malignancy may be through promoting survival of glioma stem cells. Knockdown of SEMA3C was able to reduce sphere formation and inhibit proliferation of glioma stem cells as well as impair tumor formation in intracranial xenografted glioma stem cells in mice. Furthermore, this inhibition was specific to the glioma stem cells, and spared the tumor cells without stem characteristics as well as normal neural progenitor cells [21]. These studies provide preliminary evidence that SEMA3C inhibition may be a safe and effective target to inhibit glioblastoma growth given the minimal toxic effect on normal brain tissue. 

While most of the associations between SEMA3C and brain malignancies has been identified in gliomas, SEMA3C has also been implicated in neuroblastoma. Neuroblastoma is a malignancy presenting in the pediatric population with very high metastatic potential. Interestingly, in this context SEMA3C plays a role in maintaining the cohesiveness of the tumor and elevated SEMA3C expression results in decreased metastatic dissemination [40,117]. Elucidation of the differing molecular targets and signalling pathways underlying the seemingly contradictory roles of SEMA3C would be very valuable in future evaluations of which cancers would benefit most from SEMA3C inhibitors while avoiding harm in others.

#### 4.2.3. Breast Cancer

Breast cancer is the most prevalent cancer in females [65]. One of the most prominent classifications of breast cancer tumors is whether they express the estrogen receptor (ER), progesterone receptor (PR), or human epidermal growth factor receptor 2 (HER2). Identifying the subtype of breast cancer has important implications for which targeted therapies can be used in treatment. Of these classes, the lowest five-year survival rates were in triple negative and ER-/PR-/HER2+ cancers which account for about 20% of breast cancers [118]. Triple-negative breast cancer in particular suffers from a lack of available targeted therapies, though there is recent interest in identifying alternate targets [119].

SEMA3C expression is increased in neoplastic and cancerous tissue compared to normal epithelial tissue and correlates with tumor grade and degree of angiogenesis [120,121]. SEMA3C expression was highest in the two subtypes with the poorest survival rates mentioned above [121]. Knockdown of SEMA3C in in vitro models led to decreased proliferative, migratory and invasive ability [16,120,122]. Accordingly, SEMA3C inhibition could improve breast cancer survival by reducing tumor burden and metastatic potential and may represent a novel targeted therapy for triple negative breast cancer. 

#### 4.2.4. Gastric Cancer

Gastric cancer is the third leading cause of cancer deaths and prognosis is relatively poor after diagnosis, particularly in cases diagnosed in advanced stages [123]. Consequently, there currently exists an unmet medical need for more effective treatments for gastric cancers. While chemotherapy remains first-line treatment for unresectable gastric tumors, adjuvant molecular-targeted therapies may improve patient outcomes. 

In gastric cancer, SEMA3C is highly expressed in neoplastic tissue compared to normal surrounding tissue, and knockdown of SEMA3C suppresses primary gastric tumor growth as well as metastasis to the liver and reduced microvessel density. The authors concluded that SEMA3C drives gastric cancer progression through angiogenesis [15]. While this presents SEMA3C as a possible molecular target, current evidence from clinical trials of the benefit of anti-angiogenic drugs is unconvincing. As such, further research will be required to determine whether SEMA3C inhibitors would be a suitable adjuvant therapy in this setting [124,125].

#### 4.2.5. Other Cancers

SEMA3C expression has been correlated with worse prognosis in other cancers, but the relative contribution of SEMA3C signaling for these cancers remains to be determined. Higher expression of SEMA3C in hepatocellular carcinoma is associated with larger tumor size and lower survival [22]. In lung cancer, increased SEMA3C expression is a marker of EMT [14,126]. In ovarian cancer, SEMA3C is associated with poor prognosis and transfection of SEMA3C into ovarian cancer cells confers resistance to cisplatin [12,13]. Clarifying the roles of SEMA3C and the associated downstream pathways in these and other cancers may broaden the relevance of SEMA3C inhibitors as therapeutic agents.

## 5. Potential Molecular Approaches in the Inhibition of SEMA3C as a Cancer Therapy 

SEMA3C is amenable to inhibition by a variety of existing classes of therapeutic agents. These include biologics, inhibitory small molecules, neutralizing monoclonal antibodies (mAbs), and antisense oligonucleotides (ASOs). Several agents which have been designed to target semaphorins and neuropilins have previously been reviewed [127]. Two examples of inhibitors of semaphorins and their receptors with potential anti-neoplastic activities that are currently in early phase clinical trials include monoclonal antibodies targeting SEMA4D (e.g., vx15/2503; [128]), and NRP1 (e.g., MNRP1685A, a monoclonal antibody that targets the VEGF binding domain of NRP1; [129,130]). This discussion will highlight treatment considerations as well as recent progress in drug development in the context of SEMA3C.

### 5.1. Biologics

Biologics have a high regulatory approval rating as therapeutics. We recently developed a decoy protein constituting the sema domain of Plexin B1 that abrogates SEMA3C signalling [39]. This recombinant molecule attenuates RTK signalling and delays prostate cancer growth. Biologics have also been developed for the suppression of SEMA3E signalling [131]. In this work, a soluble Plexin D1 ligand trap for SEMA3E diminished tumor growth and metastasis in breast cancer. The major advantage of biologics over other therapeutic agents is high specificity. This generally leads to relatively mild side-effect profiles. 

### 5.2. Small Molecule Drugs

The design of small molecule drugs against semaphorins has been relatively unexplored, particularly in treatment of cancer. Xanthofluvin and vinaxanthone are two small molecule inhibitors of SEMA3A but these have been studied in the context of axonal regeneration [132,133]. Nevertheless, preliminary studies indicate that blockade of SEMA3C signalling using small molecules holds some promise. Using in silico pipelines and in vitro validation, we recently identified molecular probes for SEMA3C which disrupt SEMA3C’s binding to NRP1 [134]. We also demonstrated that the small molecules which resulted in the greatest displacement of SEMA3C from NRP1 led to decreased proliferation of PCa cells and a corresponding attenuation of downstream signalling pathways including EGFR and HER2. Further studies will be required to examine whether these effects translate to reduced tumor growth in in vivo models.

Small molecule drugs are an attractive form of therapy because production is relatively inexpensive compared to the other drug classes. However, small molecules often demonstrate poorer specificity and consequently are more prone to inducing off-target effects. This is an important consideration in the design of semaphorin therapeutics given the structural similarities and shared receptors of the semaphorin family.

### 5.3. Monoclonal Antibodies

Monoclonal antibodies are widely used in cancer treatment and represent a major cornerstone in the application of precision medicine and targeted therapy. As SEMA3C is secreted, mAbs are a suitable and intuitive method of inhibition. Monoclonal antibodies are perhaps the most well-documented methodology for antagonizing semaphorin signalling and mAbs have been raised against SEMA3A and SEMA4D with the intended purpose of cancer treatment. In particular, VX15/2503, a mAb against SEMA4D, been examined in early phase clinical trials in patients with solid malignancies [135]. These studies showed high tolerability. Additional trials examining VX15/2503 in combination with immunotherapies are forthcoming (NCT03268057; NCT03425461; NCT03690986; NCT03373188). Development of anti-SEMA3C mAbs are currently underway.

Other potential approaches include developing therapeutic antibodies against SEMA3C receptors such as plexin B1, plexin D1 or NRP1 to block SEMA3C-receptor interaction or potentially developing functional antibodies that disrupt that interaction of SEMA3C receptors with associated RTKs involved in downstream signaling.

### 5.4. Antisense Oligonucleotides (ASO)

ASO therapy has been gaining interest in clinical research. Early oligonucleotide therapies were not very effective because they were degraded too quickly by nucleases within the body. However, recent technological advancements have allowed for modifications of the oligonucleotide backbone as well as improved delivery methods such as nanoparticles to mitigate those issues. Several oligonucleotides have now been approved for treatment of diseases such as familial hypercholesterolemia and age-related macular degeneration, with many more currently undergoing clinical trials [136,137]. 

One advantage of using ASO to target SEMA3C is the ability to prevent SEMA3C production by targeting SEMA3C mRNA. SEMA3C ASOs could then be used in combination with other agents targeting SEMA3C protein. However, clinical studies involving ASO as anti-cancer therapeutics are sparse, and ongoing research will be required to answer questions about potential toxicities and pharmacodynamics as well as continued refinement of ASOs to improve stability and uptake into cells. Another potential advantage of ASOs relates to the fact there exist numerous semaphorin family members which exhibit structural similarity. ASO inhibitors may allow sequence specificity to discriminate among members of related proteins.

A variety of anti-SEMA3C inhibitors are currently being developed and are undergoing preclinical studies. Inhibitors of SEMA3C could also be combined with other cancer therapeutics including RTK inhibitors, taxanes, and antagonists of the AR axis in the case of PCa, to assess the possibility of synergy. Careful consideration of the efficacy and safety profiles of anti-SEMA3C agents will be imperative to guide which treatments will be appropriate to transition into clinical use. 

## 6. Perspective

A comprehensive overview of therapies directed against the semaphorins and their receptors has been compiled previously [5]. Notably absent are inhibitors of SEMA3C since the significance of SEMA3C in cancer is only just becoming increasingly recognized [6]. SEMA3C is an attractive therapeutic target from multiple standpoints: biologically, it fills tumorigenic roles and correlates with intensified disease state while its extracellular localization renders it readily accessible by pharmacological agents. Its predicted limited function in adults also makes SEMA3C a convenient target. However, the clinical space of anti-SEMA3C therapies is largely uncharted. We now discuss topics that will need to be carefully considered when developing inhibitors of SEMA3C.

As with other semaphorins, SEMA3C’s function is regulated by post-translational modifications, including proteolytic cleavage by furin and a disintegrin and metalloproteinase with thrombospondin motifs 1 (ADAMTS1) and by glycosylation [1,16,138]. Indeed, full-length SEMA3C has contrasting function relative to its proteolytically-processed counterpart [139]. As such, functional discrepancies reported across different studies may simply reflect differences in the relative abundance of full-length versus truncated forms of SEMA3C. This nuance indicates that under any given biological context not only are levels of SEMA3C relevant but so too are the levels and repertoire of enzymes that modify SEMA3C. To this end, we and others have developed cleavage-resistant forms of SEMA3C in an effort to compensate for SEMA3C truncations [16,39,139,140,141]; however, a systematic analysis correlating various forms of SEMA3C and their activities would be informative. In two recent studies, Mumblat et al. [139] and Yang et al. [140] showed that a truncated version of SEMA3C harboring a deletion of the c-terminal 13 amino acids or wild type cleavable SEMA3C, respectively, inhibited angiogenesis but did not affect cell growth. By comparison, Peacock et al. showed that SEMA3C with mutations to its processing consensus sequence 1 (PCS1) as well as R611A and R612A, promoted cancer growth and survival via receptor tyrosine kinase signalling [39]. These seemingly opposing SEMA3C-induced effects are likely due to the different forms of SEMA3C examined (e.g., non-cleavable, cleavable, or truncated forms) as well as the utilization of different cancer models and biological systems. Nevertheless, this apparent context-dependent functional dichotomy illustrates the necessity in exercising caution when ascribing SEMA3C functions.

One of the aspects complicating SEMA3C activity relates to the multiplicity of receptors utilized by SEMA3C that consequently directs the downstream signaling pathways potentially involved (Figure 4). The array of neuropilin and plexin receptors present on the cell influences what signal transduction takes place. For example, SEMA3C mediates signalling via Plexin B1 in PCa whereas SEMA3C signals through Plexin D1 in glioma. Thus, targeting SEMA3C receptor with Plexin B1 sema domain fusion proteins would be more effective in Plexin B1-dependent tumors whereas small molecule drugs, mAbs and ASO that target SEMA3C directly would act independently of receptor usage. The diverse classes of therapeutics described above which target SEMA3C through various strategies such as receptor fusion protein, small molecule drugs, mAbs and ASO could be used in combination in order to achieve total anti-SEMA3C blockade.

Additional potential caveats that should be considered related to interfering with SEMA3C receptors include the relative promiscuity of these receptors. The same receptors are often also used by other semaphorins. For example, Plexin B1 and D1 traps may also block signaling by other known ligands of these receptors. Similarly, small molecules interfering with SEMA3C-neuropilin binding may also interfere with association of neuropilins with other semaphorins. 

Like other targeted therapies, patient stratification may be of benefit to achieving meaningful and durable clinical responses to SEMA3C inhibitors in patients. While SEMA3C is not prone to heavy mutational burden [108,109], SEMA3C-high patients – who are presumably those who would most likely benefit from treatment with SEMA3C inhibitors, should be identified. Immunohistochemistry would be a suitable approach in this regard; however, as SEMA3C is a secreted protein, it also stands to reason that tumor-shed SEMA3C would be present in blood or bodily fluids where it would be detectable in enzyme-linked immunosorbent assay (ELISA)-based biochemical assays. Further to this, considering SEMA3C levels correspond with the malignant phenotype, cancer stage, cancer grade, and other important clinical parameters in multiple cancer types [13,19,21,23,39,111,120], it is possible that circulating SEMA3C status may represent a diagnostic or prognostic tool. It remains to be seen whether such approaches will be feasible.

The phase of disease during which SEMA3C inhibitors should be deployed will be another critical element to consider. On the one hand, SEMA3C is theorized to drive the rare cancer stem cell population that is thought to be responsible for treatment-resistance and tumor relapse [12]. This would indicate that the opportune time to inhibit SEMA3C would be concurrently with first-line therapies in order to achieve complete cancer remission. On the other hand, SEMA3C expression increases with tumor stage implying that it fills functions related to cancer progression therein supporting administration later in disease. A broader understanding of SEMA3C’s oncogenic roles will be informative in making these decisions which will be further assisted by preclinical studies in animal models.

## 7. Conclusions

SEMA3C presents an attractive target as an anti-cancer therapy due to its cited involvement in numerous biological processes that surround cancer etiology and progression. SEMA3C has been discussed largely for its pro-tumorigenic roles in a multitude of cancers including prostate, pancreas, brain, breast, and stomach. Here we discuss the possibility of inhibiting SEMA3C as a cancer treatment modality and provide prospective molecular approaches for inhibiting SEMA3C such as biologics, small molecules, monoclonal antibodies, and antisense oligonucleotides. However, as with all new targeted therapies, a number of factors will need to be considered during development of anti-SEMA3C therapies. Chief among them will be specificity, patient stratification based on cancer genomic taxonomy, and suitable temporal deployment during disease course. It is hoped that this report stimulates scholarly pursuits into considering SEMA3C as a therapeutic target for treatment of prostate and other cancers.

## Figures and Tables

**Figure 1 ijms-20-00774-f001:**
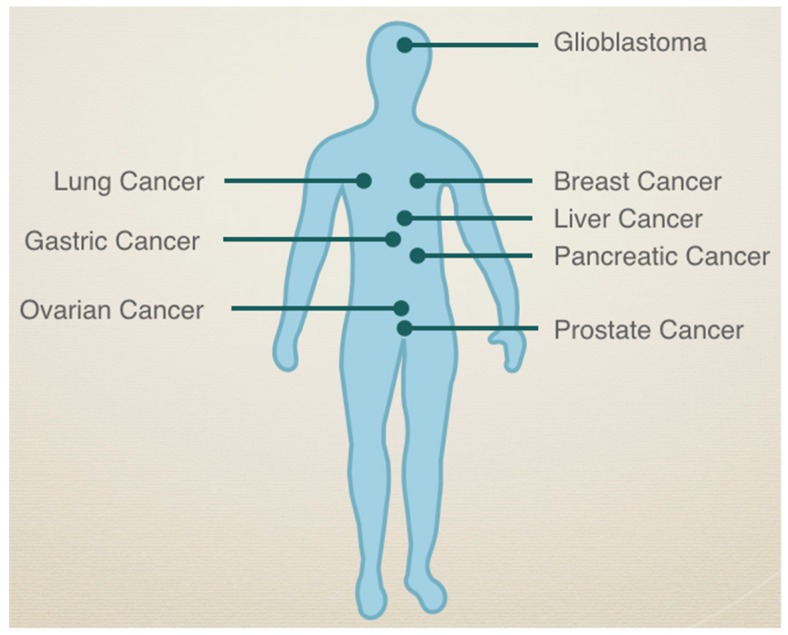
High SEMA3C expression is associated with unfavourable prognosis in multiple cancers.

**Figure 2 ijms-20-00774-f002:**
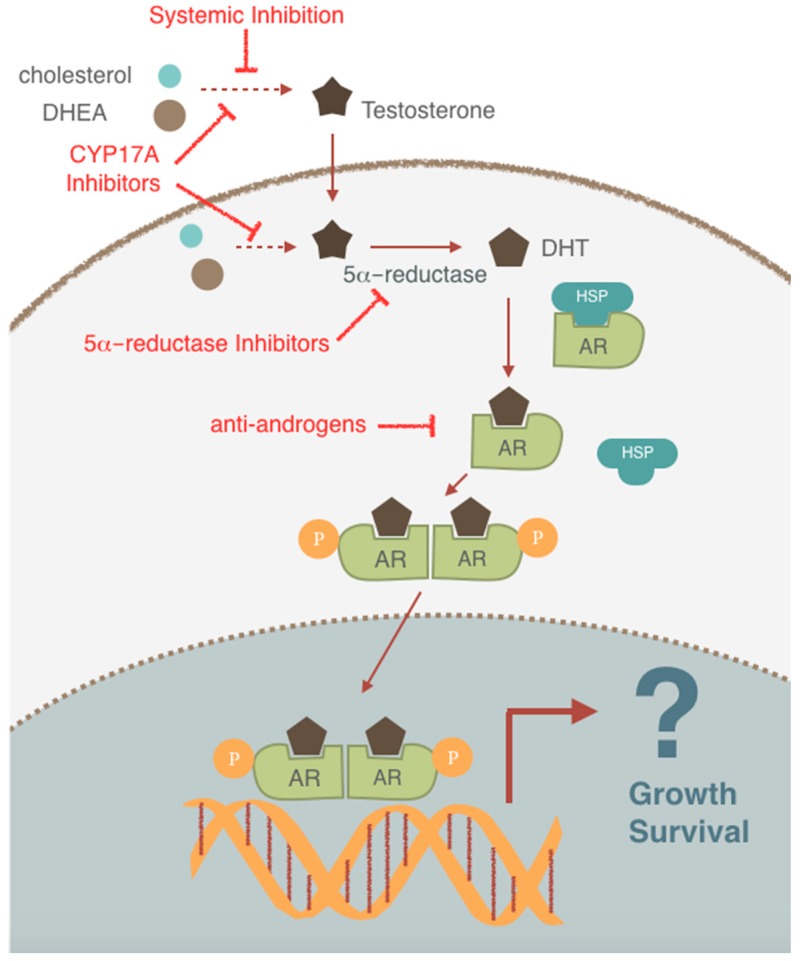
Androgen action. Testosterone circulates in the blood, enters prostate cells and is converted to dihydrotestosterone (DHT) by the enzyme 5α-reductase. Binding of DHT to the androgen receptor (AR) induces dissociation from heat-shock proteins (HSPs) and receptor phosphorylation. The AR is phosphorylated, dimerizes and translocates into the nucleus where it can bind to androgen-response elements in the promoter/enhancer regions of target genes. Activation (or repression) of target genes leads to biological responses including growth and survival. Drugs targeting various stages in the androgen/androgen receptor (AR) axis are shown. Dotted arrows represent multistep process, “T” arrows represents pathway inhibition.

**Figure 3 ijms-20-00774-f003:**
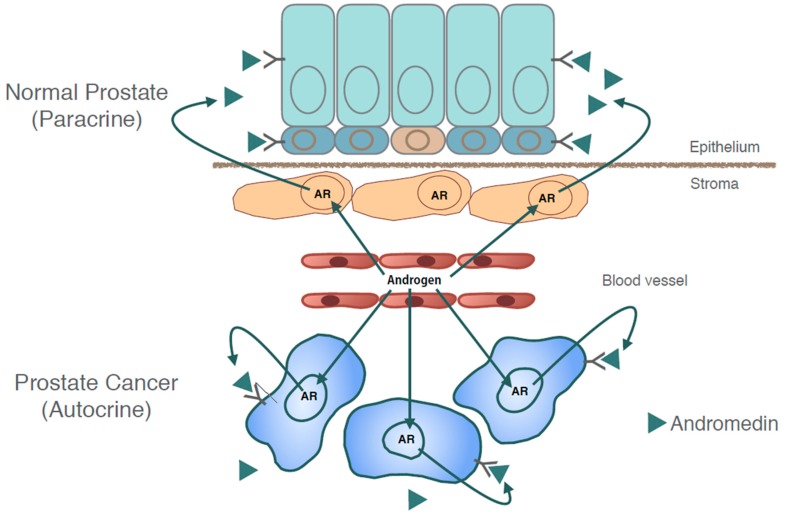
Androgen receptor signaling in normal and malignant prostate cells. In normal prostate, the growth and survival of prostate epithelium depends on secreted soluble andromedins produced by the stromal cells in an androgen-dependent manner which diffuse across the basement membrane and act on epithelial receptors (above). Malignant transformation of normal prostatic epithelial cells is associated with a switch from a paracrine to an autocrine mechanism in androgen-stimulated growth (below). Androgens diffuse from blood circulation (straight green arrows) and bind to androgen receptors in cells that leads to production of andromedins that act in an autocrine or paracrine manner (curved green arrows).

**Figure 4 ijms-20-00774-f004:**
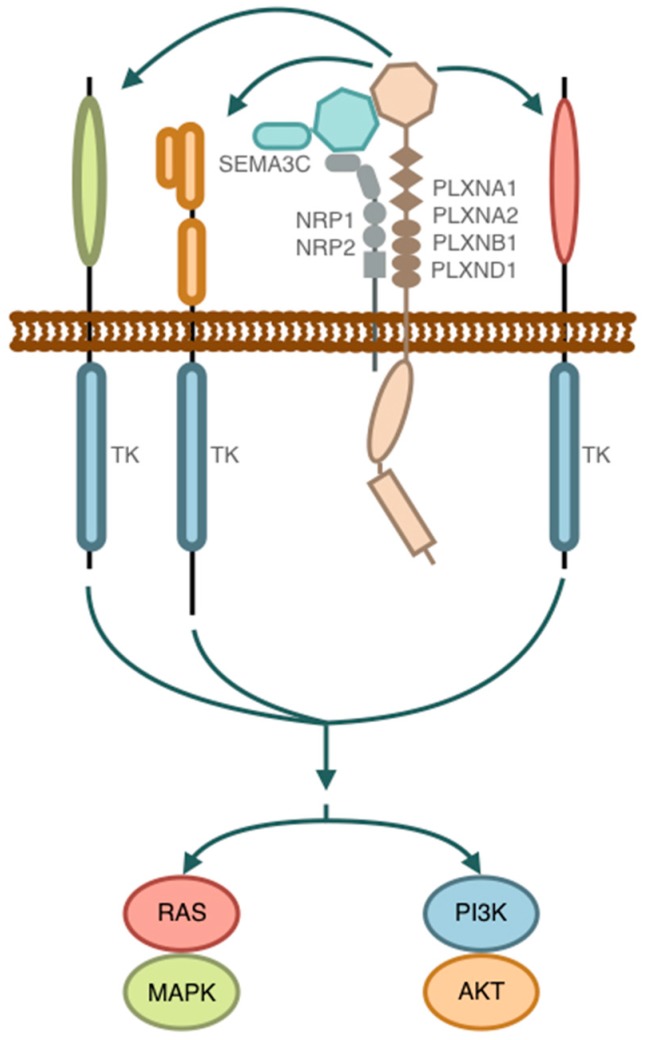
SEMA3C signalling. Binding of SEMA3C to its receptors PLXNB1 and NRP1 triggers transactivation of multiple receptor tyrosine kinases such as EGFR, HER2 and MET. Arrows indicate protein interactions and signaling pathways.

**Figure 5 ijms-20-00774-f005:**
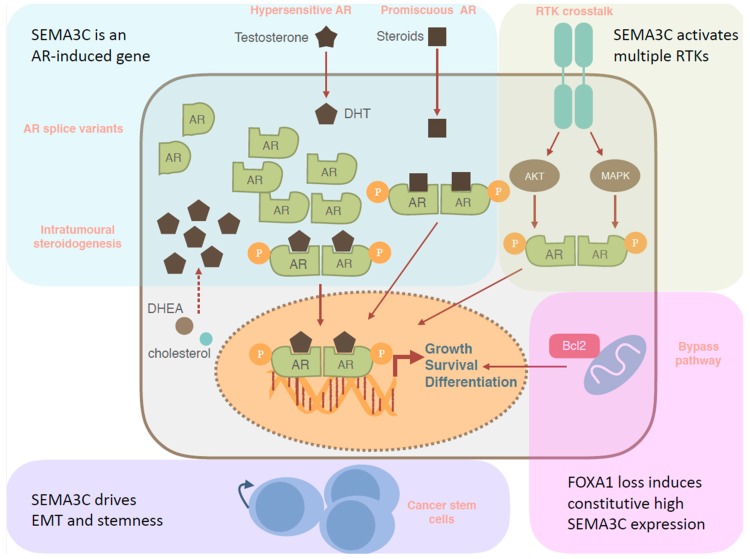
Mechanisms of CRPC development. AR signalling can be activated via AR amplification or mutation to allow signalling despite castrate levels of androgens (hypersensitive pathway), AR variants mediate AR signaling in the absence of androgens, or through intratumoral steroidogenesis de novo or from adrenal androgens. Alternatively, AR signaling may be mediated via non-androgenic steroids (promiscuous pathway), while RTK signalling cascades allow tumor cells to survive without androgens (RTK crosstalk). In the absence of AR, survival can be enhanced through cell-intrinsic pathways, such as loss of phosphatase and tensin homologue (PTEN) or upregulation of anti-apoptotic Bcl-2 proteins (bypass pathway). Androgen independent prostate cancer stem cells undergo androgen independent self-renewal, persist after ADT, and seed tumor relapse. The potential role of SEMA3C in each resistance mechanism is described as shown. AR: androgen receptor, DHEA: dehydroepiandrosterone, DHT: dihydrotestosterone, RTK: receptor tyrosine kinase. Dotted red arrows indicate multi-step process, curved blue arrows indicate cell self-renewal and red solid arrows signify signaling pathways.

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
