# Peer review of "Semaphorin 3C as a Therapeutic Target in Prostate and Other Cancers"

_ijms, 2019, doi:10.3390/ijms20030774_

Reviewer 1 Report

The authors summarized the roles of SEMA3C in tumor biology, especially prostate cancer. SEMA3C functions as an andromedin in prostate cancer to regulate prostate cancer growth and survival. The authors also provide us molecular strategies for cancer therapy by proteins, small molecules, monoclonal antibodies and antisense oligo. The new knowledge has been reported schematically, and the figures allow us to understand the pathway in the most simple and precise way. However, the information of SEMA3C inhibitors (section 5, line 376) is limited because the importance of SEMA3C for cancer therapy is just beginning to be recognized (lines 380-381). I suggest to authors show more examples of the inhibitors of other semaphorins and/or their receptors to treat for several diseases such as cancer. This point would enhance value of the manuscript.

Author Response

We agree with the reviewer and have added two examples of inhibitors of semaphorins and/or their receptors that are currently in clinical trials.  

 The following sentence and new references were added to:  line 423:

 Two examples of inhibitors of semaphorins and their receptors with potential anti-neoplastic activities that are currently in early phase clinical trials include monoclonal antibodies targeting SEMA4D (e.g. vx15/2503; (LaGanke et al., 2017)), and NRP1 (e.g. MNRP1685A;(Patnaik et al., 2014Weekes et al., 2014)).

 Reviewer 2 Report

This review article by Hui and coworkers focuses on the role of Sema3C in cancer, and on the opportunity for its targeting in therapeutic perspective. The manuscript is very well written, and provides a broad and well-illustrated survey of the state-of-the-art about Sema3C expression and function in human tumors. The Authors’ laboratory has significantly contributed to this research field, but the review also gives adequate space to all current literature.

I have only a few remarks to make.

1)      One of the aspects complicating Sema3C activity relates with the multiplicity of receptors, and consequently of the downstream signaling pathways, potentially involved. This point is briefly mentioned in the introductory part and occasionally recalled in the manuscript. Although the scope of this paper is not to focus on Sema3C signaling mechanisms, I think that a deeper understanding of this aspect will prove to be crucial for Sema3C therapeutic targeting in cancer. For this reasons, I would recommend the Authors to summarize the currently knowledge on this, maybe in a schematic Figure, or implementing Figure 4, which is presently mentioned in line 134, merely with reference to some Authors’ findings.

2)      It would seem appropriate to me to discuss further this aspect of receptor complexity also in the MS section concerning therapeutic targeting; moreover, it could perhaps be relevant to aim at interfering (e.g. by functional antibodies) with the interaction between Sema3C-receptors and RTKs involved in downstream signaling.

3)      In line 206, with reference to Sema3C-dependent regulation of cancer stem cells, it is said that “Thus, inhibitors of SEMA3C signalling represent a potential novel therapeutic approach in metastatic CRPC in which the therapeutic options are currently limited”. This is surely intriguing, but the posited link between CSC and metastasis should not be left implicit here, and should rather be explained for the general readership. Otherwise it would be unclear why to envisage here specifically Sema3C-targeting in metastatic prostate cancer vs. other conditions.

4)      In the section concerning Potential Molecular Approaches for therapeutic targeting of Sema3C, it would be appropriate to discuss the potential caveats related to interfering with its interaction with receptors, due to the relative promiscuity of these associations. For instance, Plexin B1 or Plexin D1 traps would at the same time block other known ligands of these receptors. Similarly, small molecules interfering with Sema3C-Neuropilin binding, are likely to interfere as well with the association of many ligands of these receptors (including other secreted semaphorins and VEGFs).

5)      In line  65, the text reads: “…plexin (PLXN) receptors which possess GTPase and GEF activity [28-31]….”. This is not truly accurate, since plexins are known to possess only intrinsic GAP (GTPase Activating Protein) activity, and furthermore associate both with intracellular GEF and GAP effector proteins. This should be clarified in the text.

Author Response

We thank the reviewer for his/her very insightful comments.

1) As suggested by the reviewer, we have modified Figure 4 and made the following changes to the text.

Line 81: The fact that there are multiple receptors for SEMA3C together with the fact that certain neuropilin and plexin members are shared by multiple semaphorins collectively underscore the intricacy of semaphorin signalling but also foreshadow potential challenges in targeting all of, but only, the intended semaphorin axis - the oncogenic programs of SEMA3C in the case of this report.

Line 88 The cellular activities downstream of these RTKs are broad so strong specificity of inhibitors for SEMA3C will be paramount in order to mitigate off-target effects.

Line 523 One of the aspects complicating SEMA3C activity relates to the multiplicity of receptors utilized by SEMA3C that consequently directs the downstream signaling pathways potentially involved (Figure 4).

2) we agree with the reviewer and have added the following statement in line 465:

Other potential approaches include developing therapeutic antibodies against SEMA3C receptors such as plexin B1, plexin D1 or NRP1 to block SEMA3C-receptor interaction or potentially developing functional antibodies that disrupt that interaction of SEMA3C receptors with associated RTKs involved in downstream signaling.

3) We agree with the reviewer and have modified the sentence to improve clarity and added a reference:

Line 232  Since SEMA3C promotes development of a stem-like phenotype, a state which has been proposed to mediate metastasis(Shiozawa et al., 2013), inhibitors of SEMA3C signalling represent a potential novel therapeutic approach in metastatic CRPC in which the therapeutic options are currently limited."

4) As suggested by the reviewer, we added additional caveats in the perspective section: 

line 533 Additional potential caveats that should be considered related to interfering with SEMA3C receptors include the relative promiscuity of these receptors.  The same receptors are often also used by other semaphorins.  For example Plexin B1 and D1 traps may also block signaling by other known ligands of these receptors.  Similarly, small molecules interfering with SEMA3C-neuropilin binding may also interfere with association of neuropilins with other semaphorins

5 We agree with the reviewer made appropriate changes to correct our error:

line 71 Semaphorin signalling is transduced across the plasma membrane by plexin (PLXN) receptors which possess intrinsic GAP (GTPase Activating Protein) activity(Aurandt et al., 2002Oinuma et al., 2004Swiercz et al., 2002Wang et al., 2012)and associate with intracellular GEF and GAP effector proteins but can also transactivate receptor tyrosine kinases (Casazza et al., 2010Giordano et al., 2002).